# Prognostic Role of Blood NETosis in the Progression of Head and Neck Cancer

**DOI:** 10.3390/cells8090946

**Published:** 2019-08-21

**Authors:** Anna Sophie Decker, Ekaterina Pylaeva, Alexandra Brenzel, Ilona Spyra, Freya Droege, Timon Hussain, Stephan Lang, Jadwiga Jablonska

**Affiliations:** 1Translational Oncology, Department of Otorhinolaryngology, University Hospital Essen, 45147 Essen, Germany; 2Imaging Center Essen (IMCES), University Hospital Essen, 45147 Essen, Germany

**Keywords:** head-and-neck cancer, metastasis, neutrophils, NETs, NETosis, innate immunity, G-CSF

## Abstract

Neutrophil extracellular traps (NETs) represent web-like structures consisting of externalized DNA decorated with granule proteins that are responsible for trapping and killing bacteria. However, undesirable effects of NET formation during carcinogenesis, such as metastasis support, have been described. In the present study, we evaluated the correlation between NETosis and disease progression in head and neck cancer (HNC) patients in order to establish a valid biomarker for an early detection and monitoring of HNC progression. Moreover, factors influencing NET release in HNC patients were revealed. We showed a significantly elevated vital NETosis in neutrophils isolated from early T1–T2 and N0–N2 stage patients, as compared to healthy controls. Additionally, in our experimental setting, we confirmed the involvement of tumor cells in the stimulation of NET formation. Interestingly, in advanced cancer stages (T3–4, N3) NETosis was reduced. This also correlated with the levels of granulocyte colony-stimulating factor (G-CSF) in plasma and tumor tissue. Altogether, we suggest that the elevated NETosis in blood can be used as a biomarker to detect early HNC and to predict patients at risk to develop tumor metastasis. Therapeutic disruption of NET formation may offer new roads for successful treatment of HNC patients in order to prevent metastasis.

## 1. Introduction

Head and neck cancer (HNC) is one of the most common tumor entities worldwide with an incidence of around 680,000 new cases per year [1]. The majority of HNC is histologically diagnosed as squamous cell carcinomas. At the time of diagnosis most HNC are locally advanced, which, despite multimodal treatment, often leads to disease progression or recurrence [2]. Recurrent HNC is often no longer accessible to curative therapy, turning HNC into the ninth deadliest cancer disease [1]. Although some risk factors associated with bad prognosis are well known, it still remains unclear which patients will suffer from recurrent HNC. Thus, this field is lacking a noninvasive diagnostic tool for early HNC diagnosis and for monitoring recurrent HNC progression.

Neutrophils play an important role in innate immunity, as they are the first line of defense during acute infection. Neutrophils show various strategies to protect the body from intruding microbes, fungi, and other pathogens [3]. Besides the well-known mechanisms like phagocytosis, degranulation, or production of reactive oxygen species (ROS), another antimicrobial mechanism of neutrophils has been described—neutrophil extracellular trap (NET) formation. NETs are web-like structures composed of externalized DNA decorated with histone and granule proteins. They can be released upon different stimuli, such as bacteria, cytokines, or the protein kinase C activator phorbole-12-myristat-13-acetat (PMA), resulting in pathogen killing [4]. Intensive investigations on NETs over the last decade revealed a potential involvement of the NETs in cancer progression, as it was shown that tumor cells themselves can activate neutrophils and stimulate NET formation via production of different factors like granulocyte colony-stimulating factor (G-CSF) [5]. Moreover, it was observed that released NETs, produced upon stimulation via tumor cells, are able to capture and surround distant metastatic cells and circulating tumor cells and, so, promote metastatic processes [6]. NETs were also shown to support invasion and migration of tumor cells in vitro. Degradation of NETs by DNase treatment prevented metastasis in a murine tumor model [7].

While localized tumors can be easily removed, metastatic disease remains the leading cause of death among cancer patients. Therefore, since NET formation is supposed to facilitate metastasis, we wanted to evaluate if there is a correlation between disease progression in HNC patients and NET formation by blood neutrophils. Moreover, we wanted to reveal factors that are involved in this phenomenon. This should help to reveal useful prognostic biomarkers to identify patients prone to HNC metastasis.

## 2. Materials and Methods

### 2.1. Patients

Patients with HNC (group 1, *n* = 36, blood samples for neutrophil isolation; group 2, *n* = 17, whole blood samples for SYTOX staining; group 3, *n* = 20, tumor samples) and healthy volunteers (*n* = 10) were included in the study after written local ethics committee approval; no previous chemotherapy or radiotherapy was performed. Acute inflammatory events (infectious diseases or acute phase of autoimmune disorders) 6 months prior to the enrollment were the exclusion criteria for this study. Clinico-pathological characteristics of HNC patients enrolled in this study are listed in Table 1.

### 2.2. Isolation of Blood Neutrophils

Peripheral blood was drawn into 3.8% sodium citrate anticoagulant monovettes and separated by density gradient centrifugation (Biocoll density 1.077 g/mL, Biochrome). The mononuclear cell fraction was discarded, and neutrophils were isolated by sedimentation over 1% polyvinyl alcohol, followed by hypotonic (0.2% NaCl) lysis of erythrocytes. In view of the emerging diversity of circulating neutrophil subtypes in humans, it should be noted that high-density neutrophils were investigated in this study.

The purity of the isolated neutrophils (>95%) was estimated with flow cytometry after staining with anti-CD66b (Beckman Coulter, Krefeld, Germany). Viability Dye eFluor™ 780 (eBioscience, Affymetrix, Santa Clara, CA, USA) or 4′,6-Diamidino-2-Phenylindole, Dilactate (DAPI) (BioLegend, San Diego, CA, USA) were used to determine viable cells. (Appendix A). Data were collected and analyzed with the BD FACS Canto system and BD FACS Diva 6.0 software (BD Biosciences, BD, Franklin Lakes, NJ, USA).

### 2.3. Bacteria

To stimulate NET release, *Pseudomonas aeruginosa* strain PA14 (wild-type serogroup O10 strain, cytotoxic ExoU+) was used. Bacteria were cultured in Luria–Bertani (LB) broth for 3 h to reach the early exponential phase, washed twice in phosphate buffered saline (PBS), and the optical density of a 100 μL suspension was measured in 96-well flat-bottom cell culture plates (Cellstar, Greiner Bio One International GmbH, Frickenhausen, Germany) at 600 nm using a microplate reader Synergy 2 (BioTek Instruments, Inc., Winooski, VT, USA). OD 0.4 corresponds to a bacterial density of 5 × 10^9^/mL, as determined by serial dilutions and colony-forming unit assays. Bacteria concentration was adjusted to the desired values and verified by further plating on 2% LB agar plates.

### 2.4. Head and Neck Cancer (HNC) Tumor

Tumor tissue was digested using dispase 0.2 μg/mL, collagenase A 0.2 μg/mL, and DNase I 100 μg/mL (all Sigma-Aldrich/Merck, Darmstadt, Germany) solution in DMEM (Gibco, Life Technologies/Thermo Fisher Scientific, Waltham, MA, USA) containing 10% fetal bovine serum (FCS) and 1% penicillin–streptomycin to exclude the influence of live bacteria on NET formation by neutrophils. Cells were meshed through 50 μm filters (Cell Trics, Partec, Sysmex Europe GmbH, Goerlitz, Germany), and the concentration was measured with a CASY cell counter (Innovatis, Roche Innovatis AG, Bielefeld, Germany).

The percentage of tumor-associated neutrophils from single live cells in tumor tissue was estimated in a single-cell suspension after staining with anti-CD66b (Beckman Coulter) and Viability Dye eFluor™ 780 (eBioscience, Affymetrix).

For tumor supernatant isolation, tumor weight was measured, the tissue was cut into 0.5–1 mm pieces, and the amount of medium (DMEM (Gibco, Life Technologies/Thermo Fisher Scientific) containing 10% FCS and 1% penicillin–streptomycin) was added accordingly by weight (0.6 mL per 0.02 g). The samples were incubated for 4 h at 37 °C, 5% CO_2_, and sterile medium was used as a negative control.

### 2.5. Induction of Neutrophil Extracellular Trap (NET) Formation with Pseudomonas aeruginosa

Isolated neutrophils, 25,000/well, were incubated with *P. aeruginosa* MOI 10 in a glass-bottom 96-well plate (MatTek Corporation, Ashland, MA, USA) precoated with poly-D-lysine 1 mg/mL (Sigma-Aldrich/Merck, Darmstadt, Germany) for 1 or 4 h at 37 °C and 5% CO_2_, and sterile medium was used as a negative control. As we did not observe any significant difference in NET formation in control conditions (in the absence of *P. aeruginosa)* between 1 and 4 h (see Appendix A), the control values for 4 h were used for the further analysis.

To study the effect of G-CSF on the capacity of neutrophils to form NETs, neutrophils isolated from the blood of healthy volunteers (*n* = 5) were challenged with *P. aeruginosa* MOI 10 in the absence or presence of human G-CSF (Filgrastim HEXAL, Holzkirchen, Germany) at a concentration 10 ng/mL for 4 h.

### 2.6. Induction of NET Formation by HNC Cells

The single-cell suspension of HNC tumor tissue was prepared as described above. Cell concentration was adjusted to 25,000 in 100 μL in a glass-bottom 96-welll plate (MatTek Corporation) precoated with poly-D-lysine 1 mg/mL (Sigma-Aldrich/Merck). Neutrophils isolated from peripheral blood of the same patient 25,000/100 uL were added and incubated for 4 h at 37 °C and 5% CO_2_.

### 2.7. Immunofluorescent Staining, Fluorescence Microscopy of NETs, and Analysis

Samples were fixed with paraformaldehyde (Thermo Fisher Scientific) to a final concentration 4%, permeabilized with Triton X-100 (Sigma Aldrich/Merck, Darmstadt, Germany) 0.2% containing buffer, stained with anti-histone-1 antibodies (Merck Millipore, Darmstadt, Germany) and donkey-anti-mouse-AF546 (Invitrogen, Thermo Fisher Scientific, Waltham, MA, USA) secondary antibodies, and mounted with ProLong Gold Antifade Mountant with DAPI (Invitrogen). Decondensed chromatin structures were histone-1-positive [8] and were determined as NETs. Percentage of NET-producing cells, area covered by NETs, and the average size of NETs were estimated microscopically with Zeiss AxioObserver.Z1 Inverted Microscope with ApoTome Optical Sectioning equipped (filters for DAPI and Alexa Fluor 546). Images were processed with ZEN Blue 2012 software (CarlZeiss Microscopy GmbH, Jena, Germany) and analyzed with ImageJ v1.151j8 (public domain). For obtaining binary masks of nuclei and NET structures from the two channel .czi images, the Auto-Threshold (MaxEntropy dark for nuclei and Li dark for NETs) was used. The analyze particles tool was used to quantify the number of nuclei after performing the watershed function. The same was true for the NET structures. NET area quantification was used for further analysis. All images were analyzed in a batch mode, and individual image results were created and controlled for manual correction (see Appendix A).

### 2.8. SYTOX Staining

To identify the major granulocyte subpopulations and exclude eosinophils and basophils, a DuraClone IM Granulocytes Tube set (Beckman Coulter) was used according to the manufacturer’s protocol. The SYTOX Orange (Invitrogen) nucleic acid stain was performed with whole blood from the patients within the first 2 h after obtaining, with subsequent lysis of erythrocytes with Whole Blood Lysing Reagents (Beckman Coulter). Neutrophils were gated as CD45^+^ CD15^+^ CD294^−^ as depicted in Figure 6A. SYTOX-positive cells were considered as spontaneously producing NETs. The percentage of SYTOX-positive cells from neutrophils and the mean fluorescence intensity (MFI) of SYTOX were analyzed with the CytoFLEX system (Beckman Coulter).

### 2.9. ELISA

G-CSF content in plasma and tumor supernatants was analyzed with ELISA (R&D Systems, Minnesota, U.S.) according to the manufacturer’s protocols.

### 2.10. Statistics

Statistical analysis was performed using a Kruskal–Wallis ANOVA for multiple comparisons with Bonferroni correction, a Mann–Whitney U-test for two independent samples, a Wilcoxon matched-pairs signed rank test for dependent samples, and Spearman’s R test for correlations. A *p* < 0.05 was considered significant.

### 2.11. Study Approval

The study was done in accordance to Helsinki declaration. Human studies were approved by the Ethical Committee at Medical Faculty, University Duisburg-Essen, Essen, Germany (Ethik Votum 16-7135-BO, 14.02.2017). Written informed consent was obtained from participants prior to inclusion in the study.

## 3. Results

### 3.1. Elevated Neutrophil Extracellular Trap Formation in HNC Patients

During the course of head and neck cancer, the number of neutrophils in blood and tumor progressively increases (Appendix A). This is associated with an adverse prognosis of patients [9]. However, the exact role of these neutrophils in HNC tumor progression is not clear. Therefore, we aimed to address the functionality of blood neutrophils and its changes during tumor progression. As a first approximation, we analyzed the capability of blood neutrophils, isolated from HNC patients versus healthy controls, to release NETs in the presence of bacteria. Here, the common bacteria *P. aeruginosa* was used, which colonizes the oral cavity and upper respiratory tract in pathological conditions [10]. We observed that NET formation was significantly elevated in HNC compared to healthy individuals. (Figure 1).

### 3.2. Stage of Disease Determines the Ability of Blood Neutrophils to Release NETs

There is increasing evidence that the immune system plays an important role during malignant transformation of cells and the development of cancer. As neutrophils strongly influence tumor progression and metastasis [11,12,13], we were interested if the capability to produce NETs by these cells correlated with the tumor stage. Therefore, HNC patients were divided into groups concerning their pathologically diagnosed tumor, nodus and metastasis (TNM) stage, blood neutrophils were isolated, and their NET formation capacities were assessed. We could observe that neutrophils isolated from patients in early stages of cancer (T1–T2) already spontaneously produced higher amounts of NET as compared to neutrophils from late-stage cancer patients (Figure 2A,D). After short (1 h) incubation with *Pseudomonas*, NETs were released from early-stage neutrophils at a significantly greater extent as compared to late-stage cancer patients (Figure 2B,D). This difference was gone after longer bacterial stimulation (4 h) (Figure 2C,D), implying that early-stage neutrophils can react faster on the stimulus than late-stage neutrophils. Possibly, neutrophils from early HNC stages are already preactivated in blood, while late-stage HNC neutrophils are not, therefore needing longer activation to release NETs (Appendix A).

Next, we were interested in how NET formation correlated with the stage of lymph node metastasis. Similar to T-stage groups, the early-stage patients (N0–N1) released NETs spontaneously and after only short stimulation with bacteria, while late-stage patient neutrophils needed at least 4 h of stimulation with bacteria to release NETs (Figure 3). Such late/classical NETosis was comparable between early and late stages of HNC (Figure 3C). Here again, spontaneous NETosis was elevated in early stage patients (Figure 3A).

### 3.3. Formation of NETs Correlates with Serum Levels of G-CSF in HNC Patients

To evaluate the possible mechanism responsible for tumor-mediated activation of NETosis by circulating blood neutrophils, we checked the plasma of HNC patients for tumor-derived factors, and we correlated NET production with the expression of these factors. A significant, positive correlation was observed between serum levels of G-CSF and the percentage of NETs producing neutrophils (Figure 4A).

### 3.4. G-CSF Stimulates NET Formation in Neutrophils

As the level of NETosis seems to correlate with plasma content of G-CSF in HNC patients, we wanted to confirm if G-CSF can indeed stimulate NET formation by neutrophils in our experimental conditions. We stimulated NET formation in neutrophils, as described above, and treated them additionally with G-CSF. The size of the produced NETs was assessed. We could indeed observe that G-CSF significantly stimulated NET formation (Figure 4B,C).

### 3.5. Tumor Cells Stimulate NET Release by Blood Neutrophils

Tumor cells produce multiple cytokines and growth factors that may influence neutrophil activation and functions, such as phagocytosis, degranulation, production of reactive oxygen species (ROS), or NETosis. One such factor is G-CSF [10]. To evaluate if tumor cells were able to directly stimulate neutrophil NETosis, we isolated blood neutrophils and tumor cells from the same patient and incubated them at a 1:1 proportion for 4 h in glass-bottom slides. To get single tumor cells, tissue digestion of the patient-derived tumor tissue was performed. After this, we stained DNA and histones to visualize NETs. We could observe that tumor cells indeed strongly stimulated NET release from neutrophils without the need of any additional stimuli (Figure 5).

### 3.6. SYTOX Staining of NETs in Blood as a Potential Biomarker that Differentiates Patients Prone to Disease Progression and Metastasis

The ability to release NETs is significantly elevated in patients with an early stage of disease, and late metastatic stages show a NET release comparable to healthy patients. Therefore, we decided to test if simple SYTOX staining of blood neutrophils could provide a useful prognostic tool for HNC patients. We performed SYTOX Orange staining in the DuraClone IM Granulocytes Tube set (Beckman Coulter) and could observe an elevated percentage of SYTOX^+^ neutrophils in the blood of I–III stage HNC patients, as compared to stage IV (Figure 6B). As the percentage of SYTOX^+^ neutrophils in the blood of early-stage patients was elevated, we wanted to assess if the amount of released NETs changed in different HNC stages. Indeed, we could observe significantly elevated NET amounts that were released by blood neutrophils from early HNC stages in comparison to late stages (Figure 6C).

## 4. Discussion

Neutrophils play a key role in cancer development and progression. This was proven in multiple animal tumor models [12,14,15] and clinical observations [9,16]. While numerous neutrophilic functions are anti-tumoral [17,18], neutrophils can also support tumor development and progression [19,20,21]. A recently discovered mechanism of anti-bacterial defense, NETosis, was primarily considered to have cytotoxic anti-tumor properties, as NET-derived DNA, histones, and granule proteins are described to be strongly cytotoxic [22,23]. Nevertheless, the ability of such structures to trap tumor cells and attach them to the endothelium, thus supporting their invasion into distant organs, was also demonstrated [6,19].

In this manuscript we focused our attention on the ability of blood neutrophils to form NETs in relation to the stage of HNC disease and to the concentration of plasma G-CSF. The results of the conducted experiments indicate that formation of neutrophil extracellular traps is generally elevated in HNC patients in comparison to healthy individuals. HNC patients showed bigger absolute sizes of NETs after one and after four hours, and spontaneous NET formation in these patients is strongly elevated. This probably is due to the activation of neutrophils with tumor-derived factors, such as G-CSF. Elevated sizes of NETs allow more efficient trapping of circulating tumor cells, thus supporting their spread.

We demonstrated that, particularly in the early stages of disease (T1–2, N0–2), HNC patients showed higher NET formation by blood neutrophils. This was particularly visible for early NET release (one hour of co-incubation with bacteria), implicating significant preactivation of neutrophils by the ongoing disease in early stage patients. Neutrophils isolated from late-stage HNC patients needed longer bacterial stimulation (4 h) in order to produce comparable levels of NETs. This delayed time to react on a stimulus implicates the lack of initial prestimulation in blood in such late-stage patients. The strong spontaneous NET formation in early-stage patients may contribute to their high risk for developing metastases. A similar phenomenon has been described for oral squamous cell carcinoma, when the most prominent NET formation in response to LPS and IL-17 stimuli was detected in the early stages of the disease [24].

Several mechanisms are involved in NET formation. In light of this, classical ROS-dependent and early/rapid ROS-independent ways of NET formation were described [25]. Rapid NET formation is claimed to be “vital” as the phagocytotic and migratory capacity of such neutrophils is preserved. Classic NET release seems to be suicidal, as the neutrophil dies during this process [26]. Here, we observed more rapid NET formation in early stages of HNC and a correlation between early NET release and G-CSF in plasma, while in the late stages of HNC, neutrophils reacted slower and reached high levels of NET formation after four hours of stimulation. The mechanisms responsible for this phenomenon require further investigation, but one can speculate that early-stage HNC neutrophils are more prone to vital NETosis due to the stimulation with tumor-derived factors or circulating tumor cells, while in the late stages the classical ‘suicidal’ NETosis dominates neutrophil responses. We expect that early vital NETosis can be more supportive for metastatic processes than late suicidal, due to the ability of such neutrophils to remain motile and potentially to migrate into pre-metastatic organs [27,28]. In the present study, the group of HNC patients was significantly older than healthy controls, which may influence NET formation. Nevertheless, literature data has proven that NET release by neutrophils declines with age [29]. Therefore, the observed increase of NET detected in the early stages of HNC is not influenced by the age differences between healthy and HNC individuals.

Because various types of tumors produce several cytokines (IL-6, IL-8, TNFα) and growth factors (G-CSF), which are known to stimulate the activity of neutrophils [30,31,32], we aimed to investigate the influence of tumor-derived G-CSF on NET formation by the neutrophils isolated from the blood of HNC patients. We observed that the levels of G-CSF positively correlated with the number of NET-producing neutrophils. The increased NET formation in mice bearing G-CSF-producing tumors was reported previously [5]. To confirm that elevated NET release observed in HNC patients is due to G-CSF, we stimulated neutrophils isolated from healthy volunteers with G-CSF in vitro. We could indeed observe a stimulated NET formation in such conditions, while G-CSF alone in the absence of bacterial stimulus did not influence NET formation. Importantly, we could also demonstrate that tumor cells themselves were able to stimulate NET release, possibly due to the high expression of G-CSF.

Monitoring the level of NETs released in the blood of HNC patients could provide a useful, noninvasive biomarker for early diagnosis and monitoring of disease progression. The number of NETs in patient blood can be easily assessed using SYTOX staining of whole blood samples instead of complicated and time-consuming procedures of neutrophil isolation, NET staining, and microscopy [33]. Using this method, we confirmed the results obtained using histological quantification, showing elevated NET release in the early stages of HNC and its decrease in late stages. Nevertheless, further investigations should be performed to increase the specificity of this method [26].

Metastasis of the tumor is the major cause of deaths from cancer, so the prevention of metastases remains one of the most important goals in the therapy of HNC. As NETs are suggested to stimulate this process, the degradation of NETs should provide a possible therapeutic target. In agreement, the treatment with DNase or neutrophil elastase was shown to inhibit adhesion of circulating tumor cells to the endothelium and to reduce development of metastasis in mice [19].

NETs not only take part in killing bacteria or supporting metastasis, but they are also involved in other pathological events like thrombosis. Elevated white blood cells, particularly neutrophils, are strongly associated with an increased risk of deep vein thrombosis and mortality in cancer patients receiving systemic chemotherapy [34]. With their three-dimensional web-like structure, it appears plausible that NETs in the blood stream can bind thrombocytes and participate in forming thrombotic material. Therefore, especially after surgeries when the risk of thromboembolic complication is the highest, NET formation should be therapeutically inhibited.

Clinical observations have proven a worse prognosis of HNC patients treated with G-CSF due to chemo- or radiotherapy-induced neutropenia [35]. At the same time, released G-CSF is associated with increased invasiveness of cancer cells themselves [36] or with diminished anti-tumor effects of chemotherapy. It also promotes re-vascularization and tumor growth [37]. This suggests the prognostic significance of G-CSF levels in patient plasma and suggests re-evaluating its use in cancer patients. Moreover, therapeutic inhibition of G-CSF-signaling pathways in cancer patients prone to metastasis should be considered [12].

## 5. Conclusions

In sum, we were able to show that NET formation in blood correlates with the progression of HNC disease. Based on this, blood NETosis may serve as a biological marker that can reveal HNC patients with a high risk of cancer progression and metastasis. Early identification of such patients should help to improve disease outcomes via earlier applications of relevant therapies. Moreover, such patients should be included into strict post-treatment observation programs.

## Figures and Tables

**Figure 1 cells-08-00946-f001:**
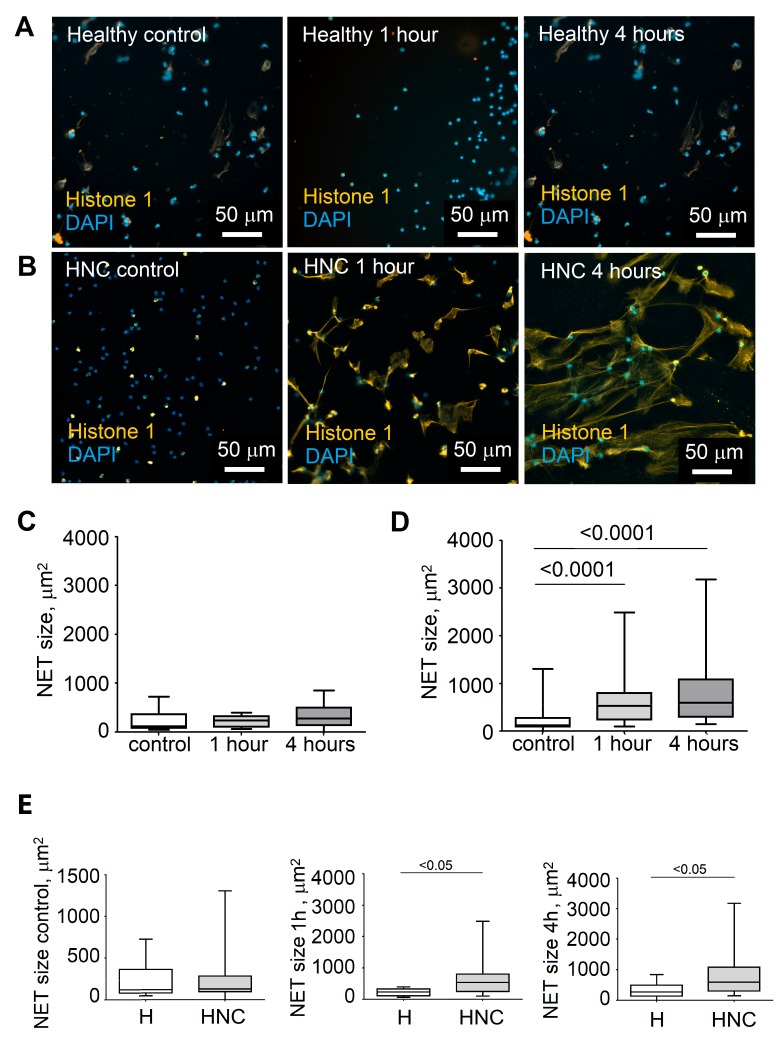
Elevated neutrophil extracellular trap formation in HNC patients. (**A**,**B**) Representative pictures showing neutrophil extracellular trap (NET) formation by healthy and HNC neutrophils. Isolated neutrophils were incubated with control medium or *Pseudomonas aeruginosa* for 1 and 4 h, fixed, and stained with DAPI (blue) and anti-histone-1 antibodies (orange); scale bar 50 μm. (**C**) Quantification of NET size in healthy individuals. (**D**) Quantification of NET size in HNC patients. (**E**) Comparison of spontaneous, early, and late NET release in healthy and HNC individuals. Blood neutrophils of healthy individuals (*n* = 10) and HNC patients (*n* = 36) were isolated and co-cultured with medium (control) or *P. aeruginosa* for 1 and 4 h. For comparisons of multiple groups, Kruskal–Wallis ANOVA with Bonferroni correction was used; for comparisons of two independent groups, Mann–Whitney U-tests were used; and for comparisons of two dependent groups, a Wilcoxon matched-pairs signed rank test was used. Data are shown as median, interquartile range, and minimal and maximal values.

**Figure 2 cells-08-00946-f002:**
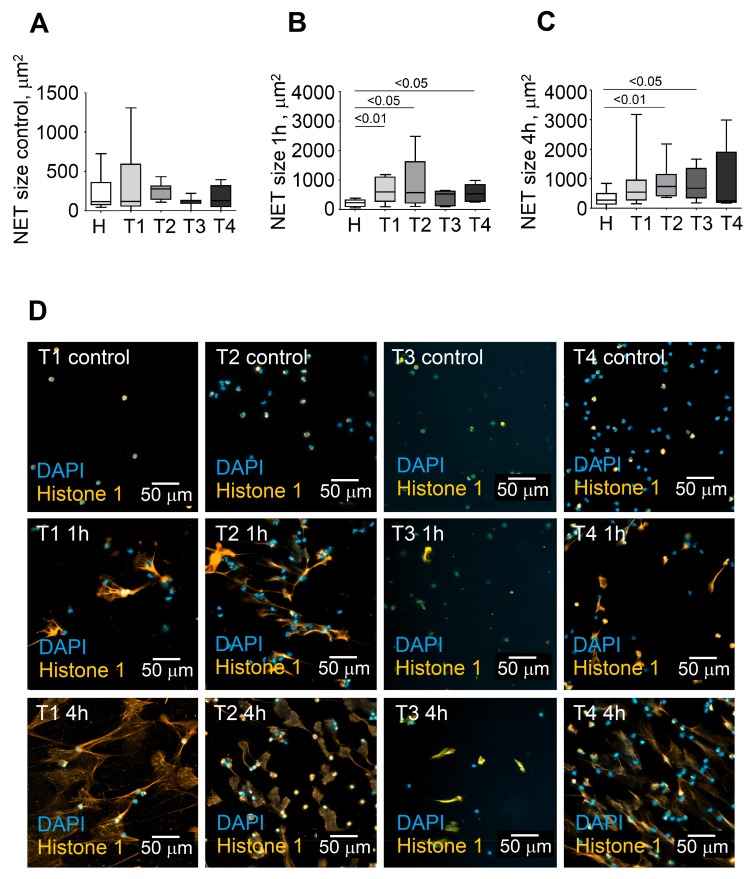
Neutrophils from early T stages of HNC release NETs significantly faster and to a greater extent. (**A**) Quantification of spontaneous NET release in healthy and HNC in different stages of disease. (**B**) Quantification of early NET release in healthy and HNC in different stages. (**C**) Quantification of late NET release in healthy and HNC in different stages. Neutrophils were isolated and stained (control, **A**), or co-cultured with *P. aeruginosa* for 1 (**B**) or for 4 (**C**) h; H—healthy. For comparisons of multiple groups, Kruskal–Wallis ANOVA with Bonferroni correction was used, and for comparisons of two independent groups, Mann–Whitney U-tests were used. Data are shown as median, interquartile range, and minimal and maximal values. (**D**) Exemplified pictures for NET formation in different HNC T stages. Isolated neutrophils were incubated with control medium or *P. aeruginosa* for 1 and 4 h, fixed, and stained with DAPI (blue) and anti-histone-1 antibodies (yellow); scale bar 50 μm.

**Figure 3 cells-08-00946-f003:**
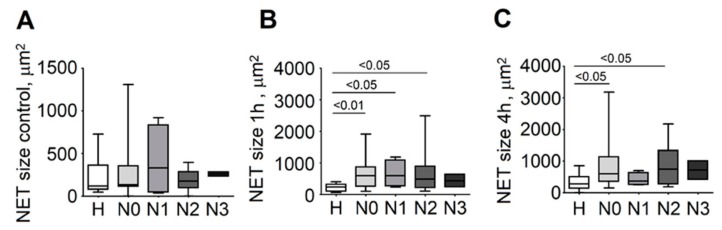
Neutrophils from early N stages of HNC release NETs significantly faster and to a greater extent. (**A**) Quantification of spontaneous NET release by different N stages of HNC neutrophils. (**B**) Quantification of early NET release. (**C**) Quantification of late NET release. Neutrophils were isolated and stained (**A**), or co-cultured with *P. aeruginosa* for 1 h (**B**) and for 4 h (**C**). H—healthy. For comparisons of multiple groups, Kruskal–Wallis ANOVA with Bonferroni correction was used, and for comparisons of two independent groups, Mann–Whitney U-tests were used. Data are shown as median, interquartile range, and minimal and maximal values.

**Figure 4 cells-08-00946-f004:**
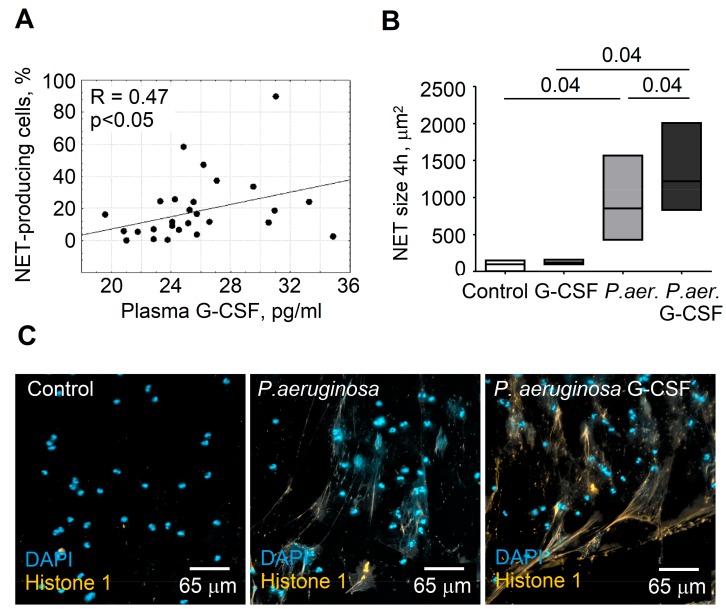
G-CSF stimulates *P. aeruginosa*-induced NET formation. (**A**) Plasma G-CSF levels correlate with the percentage of early (within the first hour of co-incubation with *P. aeruginosa*) NETs. For analysis of correlations, Spearman’s R test was used. (**B**) *P. aeruginosa*-induced NET formation of healthy donor (*n* = 5) neutrophils can be stimulated with G-CSF in vitro. Data are shown as median and min/max. For comparisons of two dependent groups, a Wilcoxon matched-pairs signed rank test was used. (**C**) Representative figures illustrating the stimulation of *P. aeruginosa*-induced NET formation by G-CSF. DAPI (blue), histone 1 (orange), scale bar 65 μm.

**Figure 5 cells-08-00946-f005:**
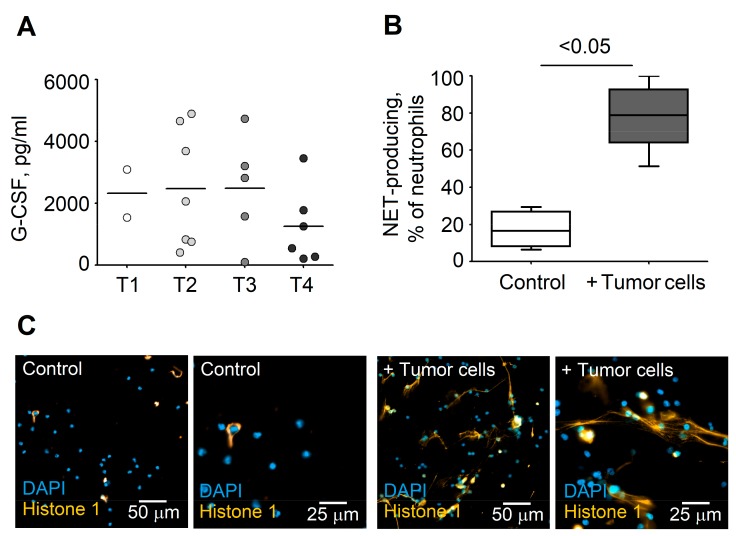
Tumor cells produce high levels of G-CSF and stimulate NETosis of blood neutrophils. (**A**) G-CSF levels in supernatants of cultured tumor tissue (normalized to sample mass). Data are shown as individual values and mean. (**B**,**C**) Elevated NET production by neutrophils co-incubated with tumor cells. Quantification of NET-positive neutrophils (**B**), and exemplified pictures showing NET release by HNC blood neutrophils (**C**). Isolated neutrophils and tumor cells of HNC patients were co-incubated at a 1:1 proportion for 4 h, fixed, and stained with DAPI (blue) and anti-histone-1 antibodies (orange); scale bars 50 and 25 μm. For comparisons of two independent groups, a Mann–Whitney U-test was used. Data are shown as median, interquartile range, and minimal and maximal values.

**Figure 6 cells-08-00946-f006:**
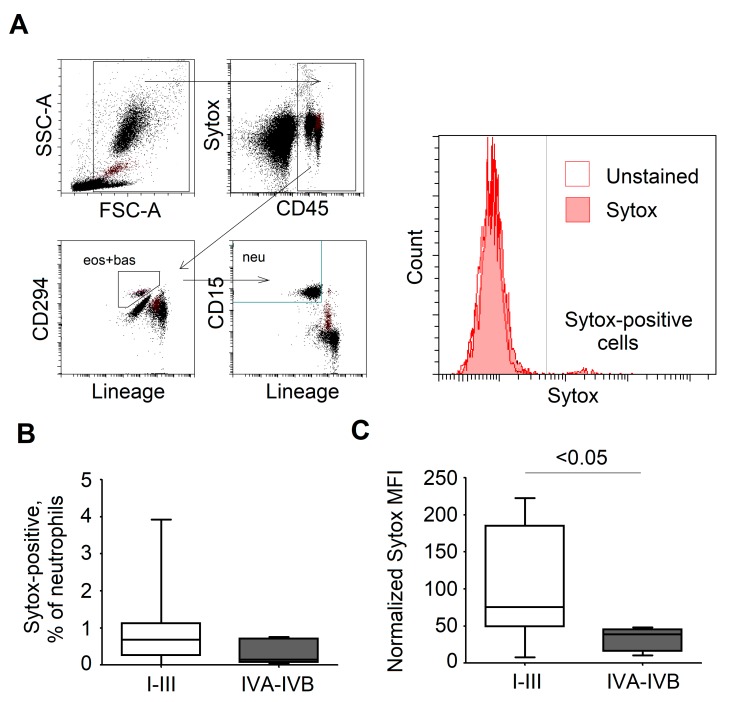
Elevated NET release in early stages of HNC demonstrated in flow cytometry using SYTOX staining. (**A**) Gating strategy for SYTOX expression on blood neutrophils. Neutrophils are gated as CD45^+^CD15^+^CD294^−^cells. NET-releasing neutrophils are determined as SYTOX-positive cells. (**B**) Elevated percentage of SYTOX-positive neutrophils in lower stages of HNC. (**C**) Elevated SYTOX expression on neutrophils (MFI) in lower stages of HNC. For comparisons of two independent groups, a Mann–Whitney U-test was used. Data are shown as median, interquartile range, and minimal and maximal values.

**Table 1 cells-08-00946-t001:** Clinico-pathological characteristics of patients enrolled in this study. HNC—head and neck cancer.

	Healthy (*n* = 10)	HNC group 1 (*n* = 36)	HNC group 2 (*n* = 17)	HNC group 3 (*n* = 20)
Age (years)	33 (22–55)	62 (28–85)	67 (52–75)	68 (52–80)
Gender, male (N, %)	5 (50%)	19 (53%)	10 (59%)	15 (75%)
Localization:	-			
larynx (N, %)		14 (39%)	6 (35.5%)	8 (40%)
oropharynx (N, %)		12 (33%)	5 (29%)	4 (20%)
other * (N, %)		10 (28%)	6 (35.5%)	8 (40%)
T Stage:	-			
1 (N, %)		14 (39%)	3 (17.5%)	2 (10%)
2 (N, %)		10 (28%)	4 (24%)	7 (35%)
3 (N, %)		7 (19%)	3 (17.5%)	5 (25%))
4 (N, %)		5 (14%)	7 (41%)	6 (30%)
N stage:	-			
0 (N, %)		18 (50%)	5 (29%)	6 (30%)
1 (N, %)		4 (11%)	3 (18%)	3 (15%)
2 (N, %)		12 (33%)	8 (47%)	11 (55%)
3 (N, %)		2 (6%)	1 (6%)	0 (0%)

* Oral cavity, glands, nasopharynx, hypopharynx.

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
