# Peer review of "Prognostic Role of Blood NETosis in the Progression of Head and Neck Cancer"

_cells, 2019, doi:10.3390/cells8090946_

Round 1
Reviewer 1 Report
Neutrophil extracellular traps (NETs) promote cancer metastasis in preclinical models, but molecular mechanisms remain unknown. In this manuscript authors performed analysis of human neutrophils isolated from patients with cancer and drew some conclusions. However, several issues should be addressed;
1. Experimental design
Why do authors choose two timepoints-1 hour and 4 hours, as well as P. aeruginosa MOI 10? Have authors ever used LPS?
2. Fig.2 and Fig.3
Whether duration of treatment could influence NET size, please perform statistical analysis via 2-tailed Mann-Whitney U test, such as comparison between 1 and 4 hours.
3. Fig.4
In vivo animal models showed that the elevation in neutrophil count, neutrophil priming and the prothrombotic phenotype could be mimicked by administration of G-CSF prior to LPS. Recent data showed G-CSF-producing tumors have also been associated with poor prognosis.
Regarding to Fig.4, Important details missed. How long were neutrophils treated with bacterial here? GM-CSF alone control group missed.
4. Fig.6
How many samples are there in per cohort? Please provide details.
Author Response
Comments and Suggestions for Authors
Neutrophil extracellular traps (NETs) promote cancer metastasis in preclinical models, but molecular mechanisms remain unknown. In this manuscript authors performed analysis of human neutrophils isolated from patients with cancer and drew some conclusions. However, several issues should be addressed .
We are grateful for all comments and remarks that improved our paper. Thank you for your time.
Experimental design
Why do authors choose two timepoints-1 hour and 4 hours, as well as P. aeruginosa MOI 10? Have authors ever used LPS?
Thank you for these questions.
The time points were chosen based on the study of Saffarzadeh et al [1], where the major difference in NETosis was observed between the early (up to 60 minutes) and late (150 minutes) NETosis. This may suggest the different mechanisms underlying early and late NET formation. As a confirmation, an extensive review of Yipp and Kubes [2] clearly described the existence of early (vital) and late (suicidal) NETosis.
MOI 10 was chosen based on the study of Young et al [3], in which this MOI was shown to be the minimal MOI able to stimulate induction of NET formation by neutrophils.
In our experimental settings, we preferred to use alive bacteria (Pseudomonas aeruginosa), as alive bacteria are more potent stimuli for neutrophils than soluble LPS. This is due to involvement of multiple mechanisms, including stimulation of other TLRs and motility sensing [4]. We feel that using bacteria provides a trigger that is more physiological than using an isolated LPS. Therefore, we have never used LPS.
Fig.2 and Fig.3
Whether duration of treatment could influence NET size, please perform statistical analysis via 2-tailed Mann-Whitney U test, such as comparison between 1 and 4 hours.
Two-tailed Wilcoxon matched-pairs signed test for dependent samples (different time points of incubation for the same patient cells) was performed as suggested (see Figure S1C).
Fig.4
In vivo animal models showed that the elevation in neutrophil count, neutrophil priming and the prothrombotic phenotype could be mimicked by administration of G-CSF prior to LPS. Recent data showed G-CSF-producing tumors have also been associated with poor prognosis.
Regarding to Fig.4, Important details missed. How long were neutrophils treated with bacterial here? G-CSF alone control group missed.
Thank you for this comment. We included G-CSF alone control into Figure 4B. We have also modified Materials and Methods chapter accordingly.
Fig.6
How many samples are there in per cohort? Please provide details.
We thank the reviewer for this important point, the details are provided in the Materials and Methods chapter.

Reviewer 2 Report
In this paper authors show that neutrophil extracellular traps (NET) formation correlates with progression of head and neck cancer. Neutrophils play very important role in innate immunity. They are the first line of defense during acute infection. Authors proposed NET as possible biological marker of a high risk of head and neck cancer progression and metastasis. According to their findings it may be possible to apply special treatments or post-treatment observation programs to patients suffering from this kind of malignancy. One of such possibility is inhibition of G-CSF signaling pathways in cancer patients prone to metastasis which was suggested in other publication. The authors very precisely analyze importance of blood NET-osis in the prognosis of head and neck tumors, particularly NET-osis as prognostic factor.
I suggest to publisj this paper.
Author Response
Reviewer 2
In this paper authors show that neutrophil extracellular traps (NET) formation correlates with progression of head and neck cancer. Neutrophils play very important role in innate immunity. They are the first line of defense during acute infection. Authors proposed NET as possible biological marker of a high risk of head and neck cancer progression and metastasis. According to their findings it may be possible to apply special treatments or post-treatment observation programs to patients suffering from this kind of malignancy. One of such possibility is inhibition of G-CSF signaling pathways in cancer patients prone to metastasis which was suggested in other publication. The authors very precisely analyze importance of blood NET-osis in the prognosis of head and neck tumors, particularly NET-osis as prognostic factor.
I suggest to publish this paper.
We thank the reviewer for his time and for the evaluation of the manuscript.
Reviewer 3 Report
The work of Dr. Decker et al. describes the characterization of neutrophils and their ability to form “neutrophils extracellular traps” (NETs) in a cohort of patients affected by head and neck cancer (HNC). The authors investigated how this ability of circulating neutrophils change depending on the staging of the disease and they found an interesting correlation with the G-CSF growth factor in inducing NETosis. The manuscript has some merit, but there are a number of issues and limitations that need to be solved by the authors before considering their paper acceptable for publication in Cells journal. The points are detailed below:
MAJOR POINTS
1) In their experiments to induce NET formation in response to bacteria exposure, the authors referred to a 'negative control' setting as the neutrophils incubated with sterile medium instead of the one containing P.aeruginosa. For how long did the authors leave the sterile medium incubated with neutrophils? 1 hour or 4 hours? In other words, can neutrophils produce NET 1 hour or 4 hours after collection if incubated with sterile medium?
2) Results section, paragraph 3.1. Did the authors verify and/or considered the hypothesis that the HNC patients could have inflammatory events before blood collection? If that happened, the neutrophils of these individuals could have been previously activated, not directly depending on the presence of the cancer.
3) Rows 169-171. The authors state that NETs formation was significantly increased by comparing HNC to healthy individuals (rows 169-171, results section and 295-298, discussion section), but in the Figure 1 there is no such comparison (just the 4h or 1h Vs. the control for each group). The statements must be corrected or a comparison between HNC and healthy volunteers reported at each fixed point (control, 1h, 4h) as the authors did for stage-classified HNC in Figure 2.
4) Figure 1 and related results. The anti-histone antibody is supposed to localize in the NET structures synthesized by neutrophils but also in the nucleus of such cells. From the representative pictures included in panels A and B of Figure 1, it is quite hard to detect the yellow signal, especially in most of the healthy cells in each panel and in the HNC control panel. It should be useful to add the splitted images. Authors are requested to discuss this point.
5) Results section, Paragraph 3.4 and discussion, rows 331-335. Is the G-CSF alone able to induce any NET formation in neutrophils? Please clarify and discuss.
6) To strengthen the message of their manuscript, the authors should also evaluate if neutrophils infiltration and NET formation in HCN tissue correlate with their findings on circulating neutrophils
MINOR POINTS
A) Row 52, please consider substituting "facilitates" with "facilitate"
B) Row 119, please consider substituting "fluorescent microscopy" with "fluorescence microscopy"
C) Row 148, please consider substituting "manufactures" with "manufacturer's"
D) Row 157, please consider substituting "Human studied" with "Human studies"
E) Rows 193, 206 and 219. Please consider substituting "extend" with "extent"
F) Row 272. Please consider substituting "percentyge" with "percentage"
Author Response
Reviewer 3
The work of Dr. Decker et al. describes the characterization of neutrophils and their ability to form “neutrophils extracellular traps” (NETs) in a cohort of patients affected by head and neck cancer (HNC). The authors investigated how this ability of circulating neutrophils change depending on the staging of the disease and they found an interesting correlation with the G-CSF growth factor in inducing NETosis. The manuscript has some merit, but there are a number of issues and limitations that need to be solved by the authors before considering their paper acceptable for publication in Cells journal. The points are detailed below:
We are grateful for the reviewer time and evaluation of our manuscript. We have addressed all the questions below.
MAJOR POINTS
1) In their experiments to induce NET formation in response to bacteria exposure, the authors referred to a 'negative control' setting as the neutrophils incubated with sterile medium instead of the one containing P. aeruginosa. For how long did the authors leave the sterile medium incubated with neutrophils? 1 hour or 4 hours? In other words, can neutrophils produce NET 1 hour or 4 hours after collection if incubated with sterile medium?
Thank you for the comment. We included now the detailed description in the Materials and Methods chapter. Briefly, yes, as described in the manuscript, HNC patient-derived neutrophils can also spontaneously produce NETs (without bacteria stimulation) probably due to the previous activation by cancer cells or cytokines/growth factors in these patients. However, we did not observe any significant differences in NET formation in control conditions between 1 and 4 hours (data are shown in the supplementary figure S1B). Therefore, for the further analysis the values for control 4 hours were used.
2) Results section, paragraph 3.1. Did the authors verify and/or considered the hypothesis that the HNC patients could have inflammatory events before blood collection? If that happened, the neutrophils of these individuals could have been previously activated, not directly depending on the presence of the cancer.
We thank the reviewer for this important note. Indeed, from our study we have excluded patients with acute inflammatory events, such as infections or acute phase of autoimmune disorders, within the last 6 months prior to admission to the hospital. We added this point into the text.
3) Rows 169-171. The authors state that NETs formation was significantly increased by comparing HNC to healthy individuals (rows 169-171, results section and 295-298, discussion section), but in the Figure 1 there is no such comparison (just the 4h or 1h Vs. the control for each group). The statements must be corrected or a comparison between HNC and healthy volunteers reported at each fixed point (control, 1h, 4h) as the authors did for stage-classified HNC in Figure 2.
As suggested by the reviewer, graphs illustrating the comparison between HNC and healthy volunteers reported at each fixed point (control, 1h, 4h) were included into Figure 1 (Fig 1E).
4) Figure 1 and related results. The anti-histone antibody is supposed to localize in the NET structures synthesized by neutrophils but also in the nucleus of such cells. From the representative pictures included in panels A and B of Figure 1, it is quite hard to detect the yellow signal, especially in most of the healthy cells in each panel and in the HNC control panel. It should be useful to add the splitted images. Authors are requested to discuss this point.
The DNA-Histone1 antibody is previously shown to detect decondensated chromatine [1], while in the nucleus of mature alive neutrophils the chromatin persists in the condensated state, which prevents binding of anti-histone antibodies. We have added splitted figures, as requested, to better visualize histone 1 staining (Figure S2).
5) Results section, Paragraph 3.4 and discussion, rows 331-335. Is the G-CSF alone able to induce any NET formation in neutrophils? Please clarify and discuss.
G-CSF alone cannot induce the NET formation by neutrophils. We have included this data into the Figure 4 and into discussion chapter.
6) To strengthen the message of their manuscript, the authors should also evaluate if neutrophils infiltration and NET formation in HCN tissue correlate with their findings on circulating neutrophils
It is indeed an interesting point. We observe that neutrophil infiltration into tumor tissue varies between the tumor stages, and that it correlates with G-CSF level in tumors. We have included this data into Figure S1D.
To detect NETs in HNC tissue is complicated due to fact that high amounts of free DNA are released from destroyed/dying cells in the tissue. Therefore, we feel that the data provided will not mirror the real NET content of the neutrophils in tumor tissue, but rather correlate with general tissue damage. Nevertheless, such data are already available [2]. Moreover, mechanisms involved in the NETosis of tumor-associated neutrophils are possibly different from blood neutrophils, as different stimuli are available in these compartments. We feel that the NETosis of tumor-associated neutrophils in tumor tissue is beyond the scope of our manuscript, as it suggests a prognostic role of blood NETosis in the progression of HNC. Nevertheless, we will surely continue on this project and address this question in our next manuscript.
MINOR POINTS
A) Row 52, please consider substituting "facilitates" with "facilitate" B) Row 119, please consider substituting "fluorescent microscopy" with "fluorescence microscopy" C) Row 148, please consider substituting "manufactures" with "manufacturer's" D) Row 157, please consider substituting "Human studied" with "Human studies" E) Rows 193, 206 and 219. Please consider substituting "extend" with "extent" F) Row 272. Please consider substituting "percentyge" with "percentage"We thank the reviewer for the careful examination of our manuscript. All critical minor points were addressed accordingly and mistakes corrected.
Reference:
Saffarzadeh M, Cabrera-Fuentes HA, Veit F, Jiang D, Scharffetter-Kochanek K, Gille C, Rooijakkers SHM, Hartl D, Preissner KT. 2014. Characterization of rapid neutrophil extracellular trap formation and its cooperation with phagocytosis in human neutrophils. Discoveries, 2(2): e19. Demers, M., S. L. Wong, K. Martinod, M. Gallant, J. E. Cabral, Y. Wang, and D. D. Wagner. 2016. Priming of neutrophils toward NETosis promotes tumor growth. Oncoimmunology. 5: e1134073.
Round 2
Reviewer 1 Report
Authors answered related question in a satisfactory manner, and quality of manuscript is improved.
Reviewer 3 Report
Authors satisfactorily answered to all the questions of the referee and they adequately discussed the points raised by the referee. The manuscript has been significantly improved by revised discussion and by the addition of novel findings. The referee thanks the authors for their work in considering the issues raised. The manuscript is now recommended for publication in Cells journal.